# Fatty Acid Synthase Confers Tamoxifen Resistance to ER+/HER2+ Breast Cancer

**DOI:** 10.3390/cancers13051132

**Published:** 2021-03-06

**Authors:** Javier A. Menendez, Adriana Papadimitropoulou, Travis Vander Steen, Elisabet Cuyàs, Bharvi P. Oza-Gajera, Sara Verdura, Ingrid Espinoza, Luciano Vellon, Inderjit Mehmi, Ruth Lupu

**Affiliations:** 1Metabolism and Cancer Group, Program Against Cancer Therapeutic Resistance (ProCURE), Catalan Institute of Oncology, 17007 Girona, Spain; ecuyas@idibgi.org (E.C.); sverdura@idibgi.org (S.V.); 2Girona Biomedical Research Institute (IDIBGI), 17190 Girona, Spain; 3Center of Basic Research, Biomedical Research Foundation of the Academy of Athens, 115 27 Athens, Greece; adapapadim@gmail.com; 4Department of Laboratory Medicine and Pathology, Division of Experimental Pathology, Mayo Clinic, Rochester, MN 55905, USA; VanderSteen.Travis@mayo.edu; 5Division of Nephrology, University of Cincinnati, Cincinnati, OH 45267, USA; ozagajbi@ucmail.uc.edu; 6School of Population Health, University of Mississippi Medical Center, Jackson, MO 39216, USA; iespinoza@umc.edu; 7Cancer Institute, School of Medicine, University of Mississippi Medical Center, Jackson, MO 39216, USA; 8Stem Cells Laboratory, Institute of Biology and Experimental Medicine (IBYME-CONICET), Buenos Aires C1428ADN, Argentina; luciano.vellon@ibyme.conicet.gov.ar; 9The Angeles Clinic & Research Institute, Cedar Sinai affiliate, Los Angeles, CA 90025, USA; inderdevta75@yahoo.com; 10Department of Biochemistry and Molecular Biology Laboratory, Mayo Clinic Minnesota, Rochester, MN 55905, USA; 11Mayo Clinic Cancer Center, Rochester, MN 55905, USA

**Keywords:** fatty acid synthase, endocrine resistance, tamoxifen, estrogen receptor, HER2

## Abstract

**Simple Summary:**

Overactivation of the human epidermal growth factor receptor 2 (HER2) is one of the main drivers of tamoxifen resistance in estrogen receptor (ER)-positive breast cancer patients. Combined targeting of HER2 and ER, however, has yielded disappointing results in the clinical setting. Therefore, other potential mechanisms for tamoxifen resistance would not be overcome by solely blocking the cross-talk between ER and HER2 at the receptor(s) level. Using cell lines, animal models, and clinical data, we provide evidence to support a critical role of fatty acid synthase (FASN)—the major site for endogenous fat synthesis—in HER2-driven tamoxifen resistance. Importantly, treatment with a FASN inhibitor impeded the estrogen-like tumor-promoting effects of tamoxifen and fully restored the anti-estrogenic activity of tamoxifen in ER+/HER2-overexpressing breast cancer xenografts. We postulate FASN as a biological determinant of HER2-driven tamoxifen resistance and FASN inhibition as a novel therapeutic approach to restore tamoxifen sensitivity in endocrine-resistant breast cancer.

**Abstract:**

The identification of clinically important molecular mechanisms driving endocrine resistance is a priority in estrogen receptor-positive (ER+) breast cancer. Although both genomic and non-genomic cross-talk between the ER and growth factor receptors such as human epidermal growth factor receptor 2 (HER2) has frequently been associated with both experimental and clinical endocrine therapy resistance, combined targeting of ER and HER2 has failed to improve overall survival in endocrine non-responsive disease. Herein, we questioned the role of fatty acid synthase (FASN), a lipogenic enzyme linked to HER2-driven breast cancer aggressiveness, in the development and maintenance of hormone-independent growth and resistance to anti-estrogens in ER/HER2-positive (ER+/HER2+) breast cancer. The stimulatory effects of estradiol on *FASN* gene promoter activity and protein expression were blunted by anti-estrogens in endocrine-responsive breast cancer cells. Conversely, an AKT/MAPK-related constitutive hyperactivation of *FASN* gene promoter activity was unaltered in response to estradiol in non-endocrine responsive ER+/HER2+ breast cancer cells, and could be further enhanced by tamoxifen. Pharmacological blockade with structurally and mechanistically unrelated FASN inhibitors fully impeded the strong stimulatory activity of tamoxifen on the soft-agar colony forming capacity—an in vitro metric of tumorigenicity—of ER+/HER2+ breast cancer cells. In vivo treatment with a FASN inhibitor completely prevented the agonistic tumor-promoting activity of tamoxifen and fully restored its estrogen antagonist properties against ER/HER2-positive xenograft tumors in mice. Functional cancer proteomic data from The Cancer Proteome Atlas (TCPA) revealed that the ER+/HER2+ subtype was the highest FASN protein expressor compared to basal-like, HER2-enriched, and ER+/HER2-negative breast cancer groups. FASN is a biological determinant of HER2-driven endocrine resistance in ER+ breast cancer. Next-generation, clinical-grade FASN inhibitors may be therapeutically relevant to countering resistance to tamoxifen in FASN-overexpressing ER+/HER2+ breast carcinomas.

## 1. Introduction

Endocrine therapies for estrogen receptor (ER) positive (ER+) breast cancer reduce cancer recurrence and death. Unfortunately, not all patients with ER+ tumors respond to endocrine therapy because of primary, de novo resistance. Moreover, those ER+ patients who initially respond can later relapse by developing secondary, acquired resistance. Indeed, 25–30% of patients with early-stage breast cancer treated with selective ER modulators (SERMs) blocking the activity of ER, such as tamoxifen, are at risk of relapsing within 2–5 years or developing resistance to the drug with ongoing treatment [1,2,3,4,5]. Accordingly, the identification of clinically important molecular mechanisms driving endocrine resistance continues to be a high priority in ER+ breast cancer.

Much of our knowledge of the molecular and biological mechanisms that govern endocrine resistance has come from studies showing a loss of ER expression, ER mutations, progesterone receptor negativity, and active cross-talk between ER and growth factor receptor (EGFR/HER2) pathways [1,2,3,4,5]. Regarding the latter, robust clinical evidence has repeatedly suggested that human epidermal growth factor receptor 2 (HER2) overexpression confers resistance to anti-estrogen therapies, even in the presence of hormone receptors [6,7,8,9,10,11,12]. At the molecular level, HER2 overexpression in ER+ breast cancer cells suffices to drive tamoxifen resistance, and elevated levels of EGFR and HER2 are detected in tamoxifen-resistant clones following long-term exposure of ER+ breast cancer cells to tamoxifen. Overall, these pre-clinical findings originally suggested that EGFR- and HER2-targeted therapies would disrupt both genomic and non-genomic cross-talk with the ER, and consequently re-sensitize breast cancer cells to tamoxifen and other endocrine therapies [13,14,15]. In the clinical setting, however, combined targeting of ER and growth factor receptors such as HER2 has failed to improve overall survival in endocrine-unresponsive disease. In this scenario, it has become increasingly clear that additional enabling factors might be co-opted during the progressive, step-wise process of endocrine therapy resistance. Among them, the pre-clinical observation that the activation of key transducers of metabolic signals—such as the PI3K/AKT/mTOR signaling pathway—confer endocrine resistance, has guided therapeutic strategies combining endocrine therapy with mTOR inhibitors to successfully manage endocrine-resistant breast cancer in clinical settings [16,17,18,19,20].

In addition to rewired kinase activity, certain metabolic shifts have been implicated in facilitating the loss of chemotherapy and targeted therapy efficacy in cancer cells. Studies from our laboratory and others revealed that a subgroup of patients with breast cancer exhibit aberrant fatty acid production mediated by activation of the fatty acid (FA) synthase (FASN) enzyme, even in the presence of abundant extracellular FAs, strongly suggesting an inherent dependency of more virulent tumors on FASN signaling [21,22,23,24,25]. FASN-related altered lipid metabolism has long been recognized to occur in tumor cell lines with acquired resistance to chemotherapeutics (e.g., doxorubicin, etoposide, cisplatin and paclitaxel) and HER2-targeted therapies (e.g., trastuzumab and lapatinib) [26,27,28,29,30,31,32,33,34,35]. Moreover, lipid-metabolic traits, including FASN activity, are emerging as key drivers not only of ER signaling, but also of resistance to endocrine therapies including tamoxifen and aromatase inhibitors [36,37,38]. Herein, we sought to clarify the role of FASN in the development and maintenance of hormone-independent growth and resistance to tamoxifen in ER/HER2-positive breast cancer. Collectively, our findings identify FASN as a novel mediator of endocrine resistance in luminal B-like, ER+/HER2+ breast carcinomas that might be exploited as potential therapeutic target and biomarker of response to tamoxifen in the clinic.

## 2. Results

### 2.1. Estradiol Stimulates FASN Protein Expression via ERα in ER+ Luminal A-Like Breast Cancer Cells

To examine whether estrogen-stimulated ER signaling contributes to the up-regulation of FASN expression in ER+ breast cancer cells, we tested whether exogenous supplementation with physiological concentrations of estradiol, a biologically active estrogen, either increased the number of cells expressing FASN and/or increased the levels of FASN expression in individual cells. Indirect fluorescence microscopy revealed that FASN protein was localized primarily in the cytosol of untreated MCF-7 cells, a well-known model of ER+ endocrine therapy-responsive breast cancer, belonging to the luminal A molecular subtype of breast cancer (i.e., ER-positive, HER2-negative, and low proliferative capacity). MCF-7 cells exposed to estradiol showed an increase in the cytoplasmic accumulation of FASN, which was notably prevented by co-exposure to tamoxifen (Figure 1A).

### 2.2. Estradiol/ERα Signaling Stimulates the FASN Gene Promoter in ER+ Luminal A-Like Breast Cancer Cells

To assess whether estradiol stimulation of FASN protein expression involved activation of FASN gene transcription, MCF-7 cells were co-transfected with a luciferase reporter gene containing a 178-bp fragment of the FASN gene promoter harboring all the elements necessary for high level expression in cancer cells, including a complex SREBP (sterol regulatory element binding protein)-binding site [39,40,41]. We then monitored changes in luciferase activity following exposure to estradiol, finding that treatment with estradiol stimulated a significant 2-to 3-fold increase in luciferase activity relative to its baseline level in untreated control cells (Figure 1B, left). Co-exposure to tamoxifen antagonized the estradiol-induced activation of the FASN luciferase reporter, demonstrating that ERα mediated the estradiol-dependent increase in promoter activity. The ability of estradiol to stimulate FASN luciferase reporter activity in an ERα-dependent manner was replicated in ER+ (luminal A-like) endocrine-responsive T47D cells (Figure 1B, right).

### 2.3. The Stimulatory Effects of Estradiol on FASN Gene Promoter Activity Involve the PI3K/AKT/SREBP Signaling Cascade

To explore the mechanism(s) by which estradiol-stimulated ERα signaling stimulates *FASN* gene expression, MCF-7 cells were co-treated with estradiol and specific inhibitors of different signal transduction pathways known to be activated in the presence of estradiol (i.e., MEK1/MEK2 → ERK1/ERK2 and PI3K → AKT). U0126, a potent inhibitor of the MEK/ERK MAPK pathway, slightly diminished basal *FASN* luciferase reporter activity in unstimulated MCF-7 cells and partially prevented estradiol-stimulated *FASN* luciferase reporter activity (Figure 1B, left). LY294002, a potent cell permeable inhibitor of the PI3K/AKT pathway, significantly decreased basal *FASN* luciferase reporter activity in unstimulated MCF-7 cells and fully prevented estradiol-stimulated *FASN* luciferase reporter activity (Figure 1B, left).

To examine whether the SREBP-binding site mediates the effects of estradiol on the activation of the *FASN* promoter, we transiently transfected MCF-7 cells with a truncated proximal *FASN* promoter lacking the region responsible for SREBP binding (FASNΔSRE). The ability of estradiol to stimulate the *FASN* luciferase reporter was completely abolished in the absence of the SREBP-binding site (Appendix A). These results strongly support earlier studies demonstrating the importance of intact SREBP-binding sites for the hormonal induction of *FASN* gene expression in cancer cells [39,40,41,42,43]. Moreover, whereas the truncated version of the *FASN* luciferase reporter remained mostly unaffected by U0126 exposure, LY294002 treatment decreased further its activity, likely by affecting the auxiliary NF-Y and Sp-1 sites in the *FASN* promoter.

### 2.4. HER2 Overexpression Suppresses the Estrogen-Inducible Phenotype of FASN Expression in ER+ Breast Cancer Cells

The overexpression of HER2 in ER-positive, anti-estrogen-sensitive breast cancer has been shown to confer resistance to tamoxifen [6,7,11]. To evaluate whether resistance of ER+ breast cancer cells to endocrine therapies could be linked, at least in part, to activation of FASN, we took advantage of an MCF-7 cell line engineered to overexpress HER2 (MCF-7/HER2-18 cells). Stable MCF-7 transfectants overexpressing HER2 are tamoxifen resistant and rapidly form xenografts in mice treated with estrogen or tamoxifen [6,11]. MCF-7/HER2-18 cells can, therefore, be viewed as a valuable in vitro model of the HER2+/luminal B breast cancer subtype (i.e., ER-positive, HER2-positive, and high proliferative capacity).

Corroborating and expanding upon earlier observations in endometrial adenocarcinoma cells exposed to estradiol [42,44] and in breast cancer cells exposed to synthetic progestins with estrogenic activity [45], immunoblotting analysis revealed that estradiol significantly increased FASN protein expression in luminal A-like MCF-7/neo breast cancer cells, which was partially prevented by co-exposure to tamoxifen. HER2-driven up-regulation of FASN protein expression in luminal B-like MCF-7/HER2-18 breast cancer cells, however, remained largely unresponsive to the regulatory effects of estradiol and/or tamoxifen (Figure 2A).

In estradiol-depleted medium, *FASN* luciferase reporter activity was two-fold higher in MCF-7/HER2-18 cells than in empty vector-transfected MCF-7/*neo* control cells (Figure 2B). However, in contrast to its effects on MCF-7/neo control cells, the addition of estradiol had no apparent effect on *FASN* luciferase reporter activity in MCF-7/HER2-18 cells (Figure 2B). Moreover, co-exposure of MCF-7/HER2-18 cells to estradiol and tamoxifen failed to modulate the hyperactivity of the *FASN* luciferase reporter (Figure 2B, left). A plausible explanation for these findings is that hyperactivation of HER2-driven signaling, particularly the PI3K/AKT pathway, switched the endocrine responsiveness of the *FASN* luciferase reporter from estradiol-inducible (in tamoxifen-responsive MCF-7/neo cells) to estradiol-unresponsive (in MCF-7/HER2-18 cells with acquired resistance to tamoxifen). In keeping with this concept, *FASN* luciferase reporter hyperactivity in MCF-7/HER2-18 cells was exquisitely sensitive to the inhibitory effects of the PI3K inhibitor LY294002 in the absence or presence of tamoxifen (Figure 2B, left). Quantitative real-time RT-PCR experiments confirmed the stronger down-regulatory effects of the PI3K inhibitor LY294002 versus the MEK inhibitor U0126 on FASN mRNA expression in tamoxifen-treated MCF-7/HER2 cells (Figure 2B, right).

### 2.5. Pharmacological Targeting of FASN Suppresses HER2-Driven Resistance to Tamoxifen In Vitro

We then explored whether the acquisition of FASN overexpression/hyperactivity might serve as part of the HER2-driven endocrine resistance program in ER+/HER2+ breast cancer cells. To do this, we quantified the impact of pharmacological FASN blockade on anchorage-independent growth as an in vitro metric of tumorigenicity in response to estradiol and/or tamoxifen. We found that HER2-negative MCF-7 and MCF-7/*neo* control cells failed to form colonies in soft-agar in the absence of estradiol, whereas MCF-7/HER2-18 cells acquired a strong anchorage-independent capacity in response to tamoxifen (Figure 3). The tamoxifen-driven, estradiol-hypersensitive tumorigenic phenotype of MCF-7/HER2-18 cells in soft-agar was fully suppressed in the presence of the small-molecule FASN inhibitor C75 (α-methylene-β-butyrolactone), a semi-synthetic derivative of the mycotoxin cerulenin [46,47]. TVB-3166, an orally-available, reversible, potent, and selective FASN inhibitor with potential clinical relevance [34,48,49], closely mimicked the ability of C75 to fully suppress HER2-driven resistance to tamoxifen in soft-agar colony formation experiments (Figure 3A). In MCF-7/*neo* control cells, however, FASN inhibitors failed to suppress estradiol-driven soft-agar colony formation in the absence of tamoxifen (Figure 3A).

To characterize whether the ability of pharmacological FASN blockade to impede the tamoxifen-enhanced tumorigenicity of ER+/HER2+ breast cancer cells involved a differential regulation of ERα expression, we evaluated the effects of FASN inhibition on the status of ERα by immunoblotting. When combined with E_2_, tamoxifen, and E_2_ plus tamoxifen, FASN blockade with graded concentrations of C75 caused a remarkable dose-dependent down-regulation of ERα that took place to a similar extent in both MCF-7/*neo* and MCF-7/HER2-18 cells (Figure 3B).

### 2.6. FASN Inhibition Reverses HER2-Mediated Resistance to Tamoxifen In Vivo

Next, we determined whether pharmacological inhibition of FASN activity might overcome HER2-determined tamoxifen resistance in animal models. Ovariectomized nude mice were transplanted subcutaneously with tamoxifen-sensitive MCF-7/neo cells and tamoxifen-refractory MCF-7/HER2-18 counterparts, and then were randomized into four groups (vehicle, tamoxifen, C75, and tamoxifen plus C75) following estrogen withdrawal (Figure 4, top panels) or with continued estrogen supplementation (Figure 4, bottom panels).

Treatment with tamoxifen drastically inhibited estrogen-stimulated tumor growth of MCF-7/neo cells; treatment with C75, however, either failed to notably modify estrogen-driven MCF-7/neo tumor growth or enhance the anti-estrogenic efficacy of tamoxifen (Figure 4, left panels). Growth of MCF-7/HER2-18 tumors was stimulated by estrogen and entirely inhibited by estrogen withdrawal, thereby indicating a bona fide estrogen dependency (Figure 4, right panels). Tamoxifen functioned like an estrogen agonist to potently enhance tumor growth in MCF-7/HER2-18 cells when single-agent tamoxifen was added to estrogen withdrawal (Figure 4, right panels). Tamoxifen-stimulated growth of MCF-7/Her2-18 tumors was strikingly suppressed in mice treated with the FASN inhibitor C75, which closely phenocopied the tumor growth inhibitory effects of estrogen deprivation. Treatment with tamoxifen failed to cause any significant reduction in MCF-7/HER2-18 tumors growing in the presence of estrogen, thereby confirming that HER2 overexpression suffices to drive tamoxifen resistance in ER-positive MCF-7/HER2-18 breast cancer cells in vivo [11]. C75 treatment drastically antagonized estrogen-stimulated MCF-7/HER2-18 tumor growth, particularly when combined with tamoxifen (Figure 4, right panels).

To further characterize an apparent antiestrogenic effect of FASN blockade in ER+/HER2+ breast carcinomas, mice bearing MCF-7/HER2-18 tumors were randomized in a second set of independent experiments to receive E_2_ supplementation, estrogen deprivation either alone or with the SERM fulvestrant (FVT; ICI 182,780), or fulvestrant with continued estrogen supplementation in the absence or presence of C75 (Figure 5). Fulvestrant, an anti-estrogen that induces destabilization and degradation of ERα, was devoid of the agonist activity of tamoxifen that is capable of stimulating the rapid membrane activities of ER in the presence of estrogen deprivation in MCF-7/HER2-18 tumors. Whereas fulvestrant did not completely antagonize the effects of E_2_ on tumor growth, adding C75 to fulvestrant therapy slightly enhanced the capacity of FVT to inhibit the growth of MCF-7/HER2-18 cells (Figure 5).

To further confirm that the estrogen antagonist properties of tamoxifen can be restored to tumor resistance after FASN blockade, we re-examined both the FASN gene regulatory effects of estradiol and/or tamoxifen and the effect of the FASN inhibitor C75 on in vivo tumorigenesis using luminal B-like BT-474 cells, which are ER+, naturally overexpress HER2, and are primarily resistant to tamoxifen, and in endocrine-unresponsive, basal-like MDA-MB-231 cells. As observed in MCF-7/HER2-18 cells, the *FASN* luciferase reporter gene was not responsive to estradiol in tamoxifen-resistant BT-474 cells. Of note, tamoxifen itself apparently behaved as a weak agonist of *FASN* reporter gene expression rather than an inhibitor in BT-474 cells irrespective of the presence of estradiol (Figure 6A).

Estrogen treatment was required to establish BT-474 xenograft tumors in mice; in such mice, treatment with tamoxifen alone failed to inhibit estrogen-stimulated tumor growth. However, when mice were treated with C75, estrogen-stimulated tumor growth was significantly inhibited by tamoxifen (Figure 6B). No effects of estradiol were observed on *FASN* luciferase reporter activity in transfected triple (ER/PR/HER2)-negative MDA-MB-231 breast cancer cells (Figure 6A). The fast-growing pattern of MDA-MB-231 tumors remained unaltered upon treatment with tamoxifen, C75, or the combination of tamoxifen plus C75 (Figure 6B).

### 2.7. ER+/HER2+ Breast Carcinomas Express High Levels of FASN Protein

The close linkage between FASN expression/activity and the acquisition of hormone independence and tamoxifen resistance prompted us to finally investigate the ability of HER2 to activate FASN expression in ER+ breast carcinomas. We took advantage of The Cancer Proteome Atlas (TCPA, http://tcpaportal.org, accessed on 30 October 2020), a cancer functional proteomics database that is a part of the Cancer Genome Atlas (TCGA) project [50,51,52]. When breast carcinomas were categorized into basal-like, HER2-enriched, and ER+ (luminal) molecular subtypes, both HER2-enriched and ER+ subgroups had significantly augmented FASN protein expression compared to basal-like breast carcinomas (Figure 7). When ER+ tumors were, in turn, sub-divided according to their HER2 status into ER+/HER2- and ER+/HER2+, the ER+/HER2+ subgroup exhibited the highest FASN protein expression compared to basal-like, HER2-enriched, and ER+/HER2-negative breast cancer groups (Figure 7). Indeed, ER+/HER2+ tumors were found to express FASN protein to levels as high as those observed in HER2-enriched tumors and significantly higher than those found in ER+/HER2-tumors (Figure 7).

## 3. Discussion

Metabolic changes in cancer are closely associated with oncogenic signaling pathways that drive not only tumor initiation and progression, but also therapeutic resistance. In the latter regard, the concept of altered lipid metabolism is gaining traction as a fundamental mechanism of resistance to chemo-, targeted-, and endocrine therapies in breast cancer [26,27,28,29,30,31,32,33,34,35,36,37,38]. We show here, that the lipogenic enzyme FASN enables HER2 signaling to circumvent the anti-cancer activity of ER-targeting SERMs such as tamoxifen (Figure 8).

Our evidence for considering FASN as a novel biological determinant of HER2-driven endocrine resistance in ER+ breast cancer is as follows: first, estradiol-inducible and tamoxifen-suppressible FASN expression in ER+ luminal A-like breast cancer cells, depends on the transient activation of PI3K/AKT and RAS/MAPK pathways; second, estradiol-unresponsive FASN expression is constitutively up-regulated in tamoxifen-resistant luminal B-like ER+/HER2+ breast cancer cells as a result of the hyperactivation of the HER2− driven PI3K/AKT (and RAS/MAPK) signaling pathways; third, pharmacological blockade of FASN activity efficiently suppresses tamoxifen-stimulated tumor growth and restores the ability of tamoxifen to antagonize estrogen in luminal B-like ER/HER2-positive breast cancer xenografts exhibiting de novo resistance to tamoxifen; fourth, FASN expression is significantly up-regulated in ER+/HER2+ breast carcinomas. These findings, altogether, strongly suggest that FASN is a co-opted signaling molecule through which activation of growth factor receptor pathways such as HER2 bypasses the inhibition of ER signaling by endocrine therapies. Thus, FASN warrants consideration as a therapeutically targetable driver of tamoxifen resistance in ER/HER2-positive breast cancer.

Although the exact reason why FASN expression negatively correlates with sensitivity to endocrine therapy in ER+/HER2+ breast carcinomas needs to be explored in depth, it is clear that that the mechanisms regulating FASN expression notably differ between molecularly distinct luminal A and luminal B-like breast cancer cells. On the one hand, FASN expression appears to be part of an estradiol-driven cellular response that leads to tamoxifen-preventable proliferation of luminal A breast cancer cells; exposure of hormone-responsive breast cancer cells to estradiol significantly up-regulates FASN protein expression in an ERα-dependent manner. Down-stream of ERα, estradiol-induced stimulation of FASN expression occurs, at least in part, at the transcriptional level, and appears to be mediated by *cis*-acting elements in the proximal *FASN* gene promoter. Up-stream of *FASN* gene expression, an estradiol/ERα-triggered cascade involving the PI3K/AKT and the RAS/MAPK pathways molecularly connects the mitogenic activity of estradiol with FASN through SREBPs. On the other hand, the linkage between ERα-related cell proliferation/survival and FASN signaling is such that tamoxifen-unresponsive FASN overexpression in endocrine-resistant ER/HER2-positive breast cancer cells might reflect the constitutive hyperactivation of the downstream (PI3K/AKT and RAS/MAPK) signaling. Moreover, FASN activation might become part of the estrogen agonist activity of tamoxifen-bound ERα but, in distinct contrast to the agonistic effects of estradiol, those of tamoxifen are dependent on HER2 signaling (Figure 8). Thus, one might expect that inactivation of FASN signaling in endocrine-resistant breast cancer with high FASN expression leads to restoration of endocrine sensitivity. Accordingly, studies with two different in vivo model systems of ER/HER2-positive tumors, namely MCF-7/HER2-18 (HER2-transfected) and BT-474 (HER2-amplified), demonstrated the restoration of the sensitivity of tamoxifen-resistant tumors to the estrogen antagonist effects of tamoxifen by inhibiting FASN activity. Tumors in patients that are similar to MCF-7/HER2-18 xenograft tumors appear to depend on both ER and HER2 pathways for survival, whereas tumors in patients similar to BT-474 xenografts may depend only on the HER pathway for survival [53]. Intriguingly, MCF-7/HER2-18 xenograft tumors were more sensitive than BT-474 tumors to the capacity of anti-FASN therapy to restore the tumor’s sensitivity to tamoxifen, an observation with potential clinical importance, as this might suggest that certain subgroups of ER-positive patients who are resistant to endocrine therapy could greatly benefit from adding a FASN blocker to combined treatments with tamoxifen and HER2-targeted therapy.

Clinically, patients belonging to the luminal-A subtype of ER+ breast cancer respond well to endocrine therapy and have a favorable prognosis. Conversely, patients of the luminal-B subtype generally show poorer responses to endocrine therapy, require adjuvant chemotherapy, and have an unfavorable prognosis [54]. Biologically, it appears that luminal B-like breast carcinomas show a broad spectrum of sensitivity to endocrine therapy, likely related to the type of up-stream signals (e.g., IGF1R, HER family members including HER2, fibroblast growth factor receptor, etc.) that converge on the PI3K/AKT/mTOR and RAS/MAPK pathways to activate down-stream cellular processes such as cellular proliferation, resistance to cell death, and metabolic reprogramming [54,55]. Thus, a therapeutically targetable predictive marker of poor response to endocrine therapy for luminal B-like breast cancer patients would be highly desirable. Our results suggest that FASN expression might correlate with the sensitivity of ER+/HER2+ luminal B-like breast cancer cells to adjuvant endocrine therapy. These findings might aid in stratifying sub-populations of high-FASN expressing ER+/HER2+ patients that could benefit from FASN inhibitors that have recently advanced to clinical studies such as the series of imidazopyridine compounds developed by Saginet Biosciences (formerly 3-V Biosciences) (TVB) including TVB-3166 and its analog TVB-2640. The latter has become the first FASN inhibitor to enter clinical trial for cancer patients and demonstrate encouraging responses in patients with non-small cell lung, ovarian, and breast cancer including HER2+ breast cancer [34,48,56]. Nonetheless, because activation of HER2 signaling pathways and proliferation genes in ER+ luminal B-like breast cancer can occur independently of HER2 overexpression via overproduction of HER ligands [54,56], it is not surprising that FASN has also been found to operate as a key enabler of tamoxifen resistance in luminal B-like ER+/HER2- breast cancer cells, exhibiting a persistent promotion of HER2/HER3 signaling due to the overexpression of the HER3 ligand heregulin [57]. In a previous study, we reported that a majority of clinically HER2-positive tumors were scored as FASN overexpressors in a series of almost 200 patients with invasive breast carcinoma [24]. Functional cancer proteomic data from The Cancer Proteome Atlas (TCPA, http://tcpaportal.org, accessed on 30 October 2020) now reveal that the ER+/HER2+ subtype was the highest FASN protein expressor compared to basal-like, HER2-enriched, and ER+/HER2-negative breast cancer groups. The molecular elucidation of FASN—a biological determinant of HER2 signaling-driven endocrine resistance in ER+ breast cancer—might accelerate new studies aimed to confirm whether FASN expression status might be employed as a useful and predictive biomarker of endocrine treatment for luminal B-like breast cancer.

We previously demonstrated that FASN inhibitors such as C75 operate as bona fide anti-estrogens, likely by alleviating AKT cross-inhibition of ERK and feeding signaling loops that boost a non-genomic ERα/MAPK cross-talk capable of transactivate ERα-driven tumor suppressive responses [36]. The estrogen-occupied ER activates both the genomic (nuclear activity that regulates gene transcription) and non-genomic (membrane activity to be intertwined with several other cellular signaling pathways). When ER is bound by tamoxifen, which normally inhibits the genomic effects of ER, the presence of a FASN blocker might restore the sensitivity to tamoxifen by preventing tamoxifen from activating HER2 through the non-genomic functions of ER while promoting a hypersensitization to estradiol-stimulated ERα transactivation that, in distinct contrast to the agonistic effects of tamoxifen, would lead to impaired cell proliferation. Further studies are needed to illuminate whether the ability of FASN blockade to eliminate HER2-ER cross-talk and restore the antitumor effects of tamoxifen involves changes in the balance of co-activators versus co-suppressors of the nuclear tamoxifen-ER complex, modifications of the phosphorylation status of ER, and/or alteration of the physical associations between HER2 and cell membrane-bound ER.

## 4. Materials and Methods

### 4.1. Materials

Improved Minimal Essential Medium (IMEM) and phenol red-free IMEM were from Biofluids (Rockville, MD, USA). Fetal bovine serum (FBS) was from Nova-Tech Inc. (Grand Island, NY, USA). Dextran-coated charcoal-treated bovine serum (CCS) was from BioSource International (Camarillo, CA, USA). Tamoxifen and estradiol were from Sigma-Chemical Co. (St. Louis, MO, USA). Fulvestrant (ICI 182,780) was a gift from AstraZeneca. C75 was purchased from Alexis Biochemicals (San Diego, CA, USA), dissolved in DMSO, and stored as a stock solution (25 mg/mL) in the dark at −20 °C until use. LY29400, a specific inhibitor of the p110 catalytic subunit of PI-3K, and the MEK1/MEK2 inhibitor U0126 were purchased from Calbiochem (San Diego, CA, USA), dissolved in DMSO, and stored as 10 mmol/L stock solutions in the dark at −20 °C until use. TVB-3166 was kindly provided by Sagimet Biosciences (formerly 3-V Biosciences, San Mateo, CA). For treatments, estradiol, tamoxifen, fulvestrant, LY294002, U0126, C75, and TVB-3166 were freshly prepared from stock solutions and diluted with growth medium. Control cells were cultured in medium containing the same concentration of DMSO (*v*/*v*) as the experimental cultures.

The primary antibody for FASN immunoblotting and immunostaining was a mouse IgG_1_ FAS monoclonal antibody (clone 23) from BD Biosciences Pharmingen (San Diego, CA, USA). An anti-β-actin goat polyclonal antibody was purchased from Santa Cruz Biotechnology (Santa Cruz, CA, USA).

### 4.2. Cell Lines and Culture Conditions

MCF-7, MCF-7/HER2-18, BT-474, and MDA-MB-231 breast cancer cell lines were grown in IMEM containing 5% (*v*/*v*)-heat-inactivated FBS and 2 mmol/L l-glutamine. Cells were maintained at 37 °C in a humidified atmosphere of 95% air and 5% CO_2_. MCF-7 cells stably overexpressing the HER2 oncogene (MCF-7/HER2-18) were kindly provided by Dr Mien-Chie Hung (University of Texas M.D. Anderson Cancer Center, Houston, TX, USA).

### 4.3. In Situ Immunofluorescent Staining of FASN

Cells were propagated in E_2_-deprived (phenol red-free) IMEM with 5% CCS for 5 days before the onset of experiments. For experiments, MCF-7 were seeded at 1 × 10^4^ cells/well in a four-well chamber slide (Nalge Nunc International, Rochester, NY, USA). After 48 h incubation with E_2_ in the absence or presence of tamoxifen, cells were washed with PBS, fixed in 4% paraformaldehyde in PBS for 10 min, permeabilized with 0.2% Triton X-100/PBS for 15 min, and stored overnight at 4 °C with 10% horse serum in PBS. The cells were washed and then incubated for 2 h with an anti-FASN mouse monoclonal antibody diluted 1:200 in 0.05% Triton X-100/PBS. After extensive washes, the cells were incubated for 45 min with FITC-conjugated anti-mouse IgG secondary antibody (Jackson ImmunoResearch Labs, West Grove, PA, USA) diluted 1:200 in 0.05% Triton X-100/PBS. The cells were washed five times with PBS and mounted with VECTASHIELD + DAPI (Vector Laboratories, Burlingame, CA, USA). As controls, cells were stained with primary or secondary antibody alone. Indirect immunofluorescence was recorded on a Zeiss microscope. Images were noise-filtered, corrected for background, and prepared using Adobe Photoshop.

### 4.4. FASN Promoter Activity

Cells growing in phenol red-free IMEM and 5% CCS for 5 days were trypsinized and re-plated in 24-well plates at a density of 50,000 cells/well. Cells were incubated for 18 h to allow for attachment and then transfected with 300 ng/well of the pGL3-Luciferase construct (Promega, Madison, WI, USA) construct containing a luciferase reporter gene driven by a 178-bp FASN promoter fragment using FuGENE 6 transfection reagent (Roche Biochemicals, Indianapolis, IN, USA). An internal control plasmid pRL-CMV (30 ng/well) was used to correct for transfection efficiency. After 18 h, transfected cells were washed and then incubated in standard conditions (5% CCS) with estradiol, ICI 182,780, U0126, LY294002 combinations as specified, ethanol or DMSO as controls. Approximately 24 h after treatments, luciferase activity was detected using a luciferase assay system (Promega) using a VICTOR^2^_™_ 1420 Multilabel Counter (Perkin Elmer, Waltham, MA, USA). The magnitude of activation in FASN promoter-luciferase-transfected cells was determined by normalization to the luciferase activity in cells co-transfected with equivalent amounts of the empty pGL3-luciferase vector lacking the FASN promoter (Ø-luciferase) and the internal control plasmid pRL-CMV, which was taken as 1.0-fold. The control value was used to calculate the relative (fold) change in transcriptional activities of FASN promoter-luciferase-transfected cells in response to treatments after normalization to pRL-CMV activity.

### 4.5. Immunoblotting Analyses of FASN and ERα

Immunoblotting assays and immunodetection of FASN and ERa were carried out as previously described [36,42] using an anti-FASN monoclonal antibody (clone 23) from BD Biosciences Pharmingen (San Diego, CA, USA) and an anti-ERα rabbit polyclonal antibody (sc-542 from Santa Cruz Biotechnology, Inc., Santa Cruz, CA, USA), respectively. An anti-β-actin goat polyclonal antibody was purchased from Santa Cruz Biotechnology, Inc. (Santa Cruz, CA, USA).

### 4.6. Soft-Agar Colony Formation Assays

The efficiency of colony formation in liquid culture was determined by monitoring anchorage-independent cell growth in soft-agar. Briefy, MCF-7, MCF-7/neo, and MCF-7/HER2-18 cells were grown in phenol red-free IMEM and 5% charcoal calf serum (CCS) for 5 days in T-75 flasks to deplete estrogen. A bottom layer of 1.5 mL (2×) phenol red-free IMEM containing 1.2% agar and 10% CCS was prepared in 6-well plates. After the bottom layer solidified, cells (20,000 cells/well) were added in a 1 mL top layer containing either estradiol (10^−9^ mol/L), the anti-estrogen 4-OH-tamoxifen (10^−7^ mol/L), C75 (7.5 μg/mL), TVB-3166 (100 nmol/L), or their combinations in 0.7% agar and 10% CCS. Plates were incubated in a humidified 5% CO_2_ incubator at 37 °C, and colonies measuring ≥50 µm were counted after 16 days using a cell colony counter (Optronix GelCount™, Abingdon, UK) after staining with nitroblue tetrazolium (Sigma-Aldrich, St. Louis, MO, USA). Assays were carried out three times in triplicate.

### 4.7. Xenograft Studies

MCF-7, MCF-7/HER2-18, BT-474, and MDA-MB-231 cells were maintained as described above. Xenografts were established by injecting 2 × 10^6^ cells subcutaneously into ovariectomized 3–4 week old athymic female nude-Doxn1^nu^ mice (Harlan Sprague Dawley, Madison, WI, USA) that had been implanted with slow-release 17β-estradiol pellets (0.72 mg, 60-day release pellets; Innovative Research of America) implanted subcutaneously around left forearm using a trocar. When tumors reached a size of approximately 100 mm^3^, mice bearing MCF-7, MCF-7/HER2-18, BT-474, and MDA-MB-231 cells were randomly allocated to continued estrogen treatment or to estrogen withdrawal (by removal of the estrogen pellets) with vehicle alone (untreated group), single-agent tamoxifen (5 mg, 60-days release pellets; Innovative Research of America), single-agent fulvestrant (5 mg/100 mL s.c. injection twice a week; Tocris Bioscience), single-agent C75 (30 mg/Kg b.w.; i.p.), or their combinations (i.e., tamoxifen plus C75 or fulvestrant plus C75) for seven weeks (*n* = 10 animals/experimental group). Tumor volume was calculated by 3D measurements using the formula: tumor volume (mm^3^) = (length × width × height)/2. Tumor volume values (mean ± S.D.) were calculated weekly for each experimental group using a Vernier caliper in a blinded manner to minimize experimental bias. Mice were euthanized at completion of the experiment or when tumors reached a volume of 1000 mm^3^, and tumor tissues were removed and maintained at −190 °C for later analyses. Animal studies were conducted in accordance with the Guide for the Care and Use of Laboratory Animals and approved by the Mayo Institutional Animal Care and Use Committee (IACUC).

### 4.8. Analyses of Clinical Data

FASN protein expression data for the TCGA—Breast Invasive Carcinoma (BRCA) study (data normalization level 4) were obtained from The Cancer Proteome Atlas (TCPA, https://tcpaportal.org/tcpa/, accessed on 30 October 2020). Only primary tumor samples from patients with both protein expression data from TCPA and available information on ER, PR (progesterone receptor), and HER2 immunohistochemical status were included in the analysis. Replicates-based normalization (RBN) employed replicate samples that are common between batches; RBN adjusts the means and standard deviation of all the antibodies in the batches so that the means and standard deviations of the replicates become the same in all the batches. Clinical data for the same TCGA dataset were obtained from The Cancer Genome Atlas (TCGA) using the cBioportal (https://www.cbioportal.org, accessed on 30 October 2020).

### 4.9. Statistical Analysis

For all experiments, at least three independent experiments were performed with *n* ≥3 replicate samples per experiment. Investigators were blinded to animal data allocation. Experiments were not randomized. Data are presented as mean ± S.D. Comparisons of means of ≥3 groups were performed by one-way ANOVA and Dunnett’s t-test for multiple comparisons using GraphPard Prism (GraphPad Software, San Diego, CA, USA). In all studies, *p*-values < 0.05 were considered to be statistically significant (denoted as *). All statistical tests were two-sided.

## 5. Conclusions

Growth factor receptors such as HER2 have frequently been associated with both experimental and clinical endocrine therapy resistance; however, combined targeting of ER and associated pathways has failed to improve overall survival in endocrine non-responsive disease. We now reveal that FASN could be involved as an intrinsic factor of the transition to endocrine resistance in ER+/HER2+ breast cancer. The ability of FASN-related cellular metabolism to confer endocrine resistance in ER/HER2+ breast cancer cells, provides strong rationale for incorporating next-generation FASN inhibitors in the therapeutic arsenal against de novo endocrine resistance in patients with ER+/HER2+ luminal B-like breast cancer.

## Figures and Tables

**Figure 1 cancers-13-01132-f001:**
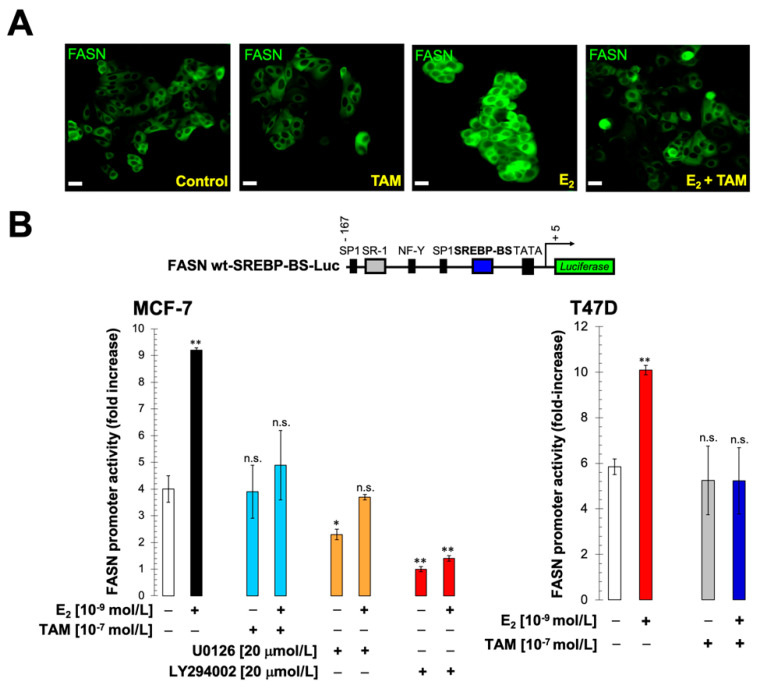
Estradiol supplementation up-regulates fatty acid (FA) synthase (FASN) protein expression in estrogen receptor-positive (ER+), luminal A-like MCF-7 breast cancer cells. (**A**) E_2_-depleted MCF-7 cells were seeded at 1 × 10^4^ cells/well in a four-well chamber slide and, after 48 h incubation with 10^−9^ mol/L E_2_ in the absence or presence of 10^−7^ mol/L tamoxifen (TAM), FASN cellular localization was evaluated after 2 h incubation with an anti-FASN mouse monoclonal antibody diluted 1:200 in 0.05% Triton X-100/PBS. After labeling, cells were extensively washed and localization of FASN was detected by indirect immunofluorescence using an FITC-conjugated anti-mouse IgG secondary antibody. A representative immunostaining analysis is shown. Similar results were obtained in 3 independent experiments. (E_2_: Estradiol; TAM: Tamoxifen; ERα: Estrogen Receptor alpha; FASN: Fatty acid synthase. Scale bar is 10 μm). (**B**) MCF-7 (left) and T47D (right) cells were transiently transfected with a plasmid containing a luciferase reporter gene driven by a 178-bp FASN promoter fragment harboring a SREBP (sterol regulatory element binding protein)-binding site, flanked by auxiliary NF-Y and Sp-1 sites (top) or with a similar construct with the SREBP-binding site deleted (bottom). The next day, cells were treated with E_2_, tamoxifen, U0126, LY294002 or their combinations, as specified. After 24 h of exposure, cells were lysed, and luciferase activity was measured. luciferase activity was expressed as relative (fold) change in transcriptional activities of FASN promoter-luciferase-transfected cells in response to treatments after normalization to pRL-CMV activity, as described in “Material and methods”. Each experimental value represents the mean fold-increase (columns) ± S.D. (bars) from at least three independent experiments in which triplicate wells were measured. * *p* < 0.05 and ** *p* < 0.005, statistically significant differences from the untreated (control) group. n.s. not significant.

**Figure 2 cancers-13-01132-f002:**
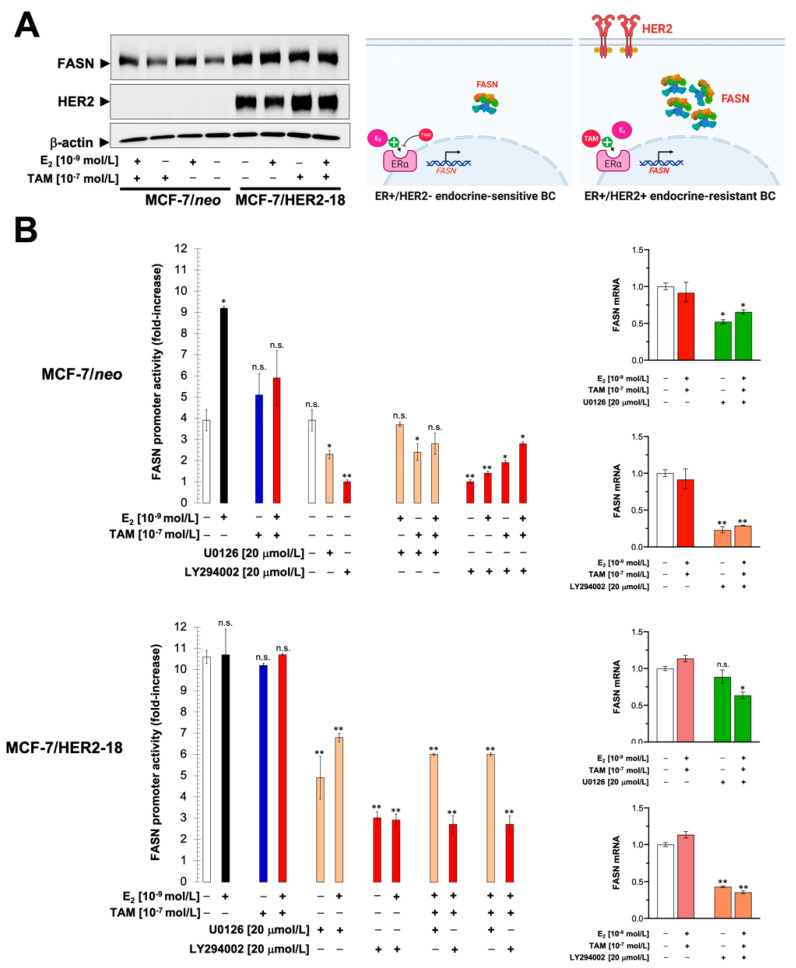
Estradiol and tamoxifen-unresponsive regulation of FASN expression in ER/HER2-positive breast cancer cells. (**A**) FASN protein expression in MCF-7/*neo* and MCF-7/HER2-18 cells cultured in the presence or absence of estradiol (E_2_), tamoxifen (TAM), or their combinations. Shown is a representative immunoblotting analysis. Similar results were obtained in 3 independent experiments. (**B)** Left. FASN promoter activity in MCF-7/neo and MCF-7/HER2-18 cells cultured in the absence or presence of estradiol (E_2_), tamoxifen (TAM), the MAPK inhibitor U0126, the PI3K inhibitor LY294002 or their combinations, was carried out as in Figure 1B. * *p* < 0.05 and ** *p* < 0.005, statistically significant differences from the untreated (control) group. *Right.* Total RNA from MCF-7/neo and MCF-7/HER2-18 cells was characterized in technical triplicates for the abundance of *FASN* mRNA relative to housekeeping genes GADPH and 18S. The transcript abundance was calculated using the delta Ct method and presented as relative quantification. * *p* < 0.05 and ** *p* < 0.005, statistically significant differences from the untreated (control) group. HER2: human epidermal growth factor receptor 2. n.s. not significant. The uncropped blots are shown in Appendix A.

**Figure 3 cancers-13-01132-f003:**
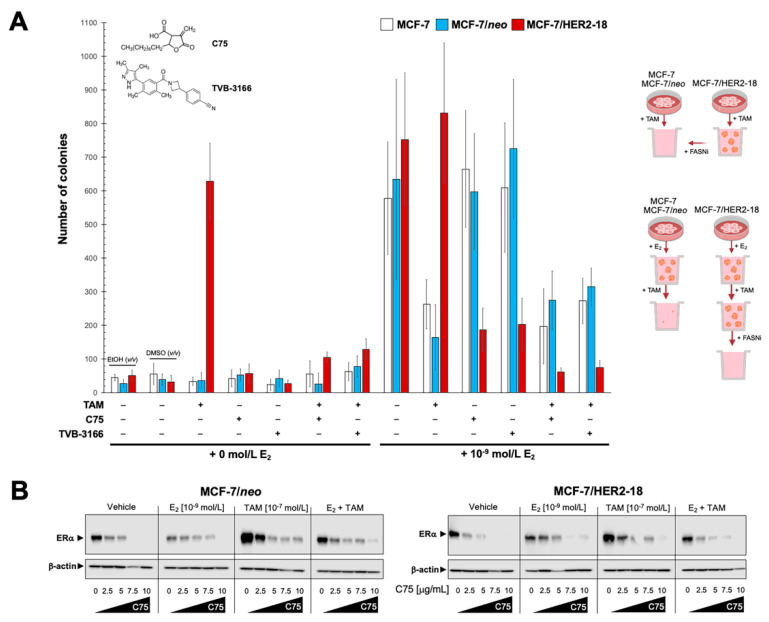
Pharmacological inhibition of FASN activity impedes the tamoxifen-driven cell growth of ER+/HER2+ breast cancer cells. (**A**) Estradiol (E_2_)-depleted cells were plated in soft agarose containing E_2_ (10^−9^ mol/L), tamoxifen (10^−7^ mol/L), C75 (7.6 µg/mL), TVB-3166 (100 nmol/L), their combinations, or ethanol (*v/v*) or DMSO (*v/v*) vehicle only for 15 days. Colony formation (≥50 μm) was assessed using a colony counter. Each experimental value represents the mean colony number (columns) ± S.D. (bars) from three separate experiments in which triplicate dishes were counted. n.s. not statistically significant. (**B**) Immunoblot analyses of ERα in MCF-7/neo and MCF-7/HER2-18 cells cultured in the absence or presence of E_2_ (10^−9^ mol/L), tamoxifen (10^−7^ mol/L), E_2_ plus tamoxifen, and/or graded concentrations of C75 were carried out, as described in “Materials and methods” using an anti-ERα rabbit polyclonal antibody. Blots were reprobed with an anti-β-actin goat polyclonal antibody to control for protein loading and transfer. Figure shows a representative immunoblot analysis. Similar results were obtained in three independent experiments. The uncropped blots are shown in Appendix A.

**Figure 4 cancers-13-01132-f004:**
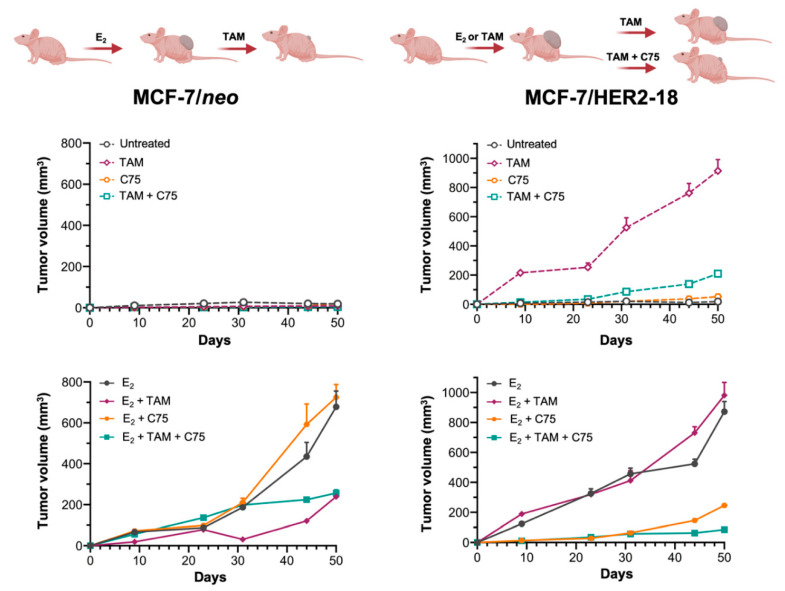
FASN inhibition suppresses tamoxifen-stimulated growth of MCF-7/HER2-18 tumors. Growth of MCF-7 and MCF-7/HER2-18 xenograft tumors in athymic female mice treated with C75 in the absence or presence of estrogen (E_2_) or tamoxifen. The maximum length for each treatment was 50 days. Results are presented as the mean tumor volume ± S.D. (*n* = 10 mice/experimental group).

**Figure 5 cancers-13-01132-f005:**
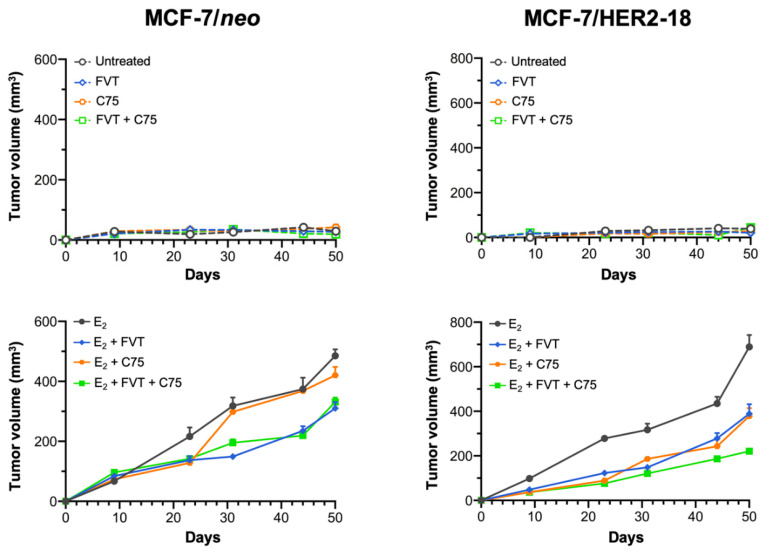
FASN inhibition enhances fulvestrant activity against MCF-7/HER2-18 tumors. Growth of MCF-7 and MCF-7/HER2-18 xenograft tumors in athymic female mice treated with C75 in the absence or presence of estrogen (E_2_) or fulvestrant (FVT). The maximum length for each treatment was 50 days. Results are presented as the mean tumor volume ± S.D. (*n* = 10 mice/experimental group).

**Figure 6 cancers-13-01132-f006:**
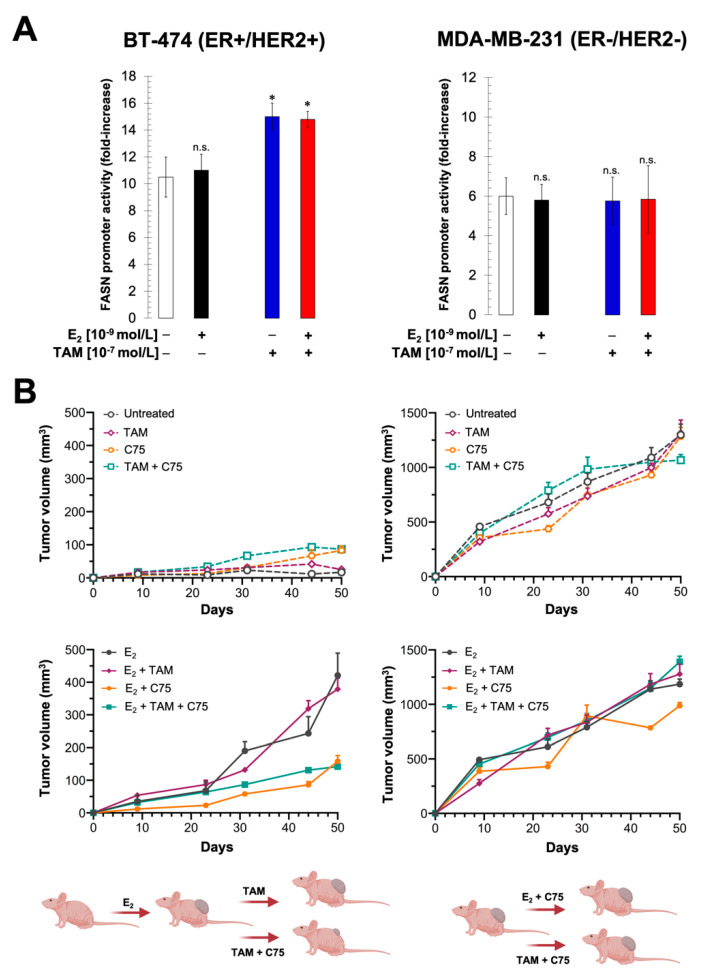
FASN inhibition suppresses primary resistance to tamoxifen of BT-474 tumors. (**A**) FASN promoter activity in BT-474 (left) and MDA-MB-231 (right) cells cultured in the presence or absence of estradiol (E_2_) tamoxifen (TAM), or the E_2_ plus TAM combination, was carried out as in Figure 1B. * *p* < 0.005; (**B**) Growth of BT-474 (left panels) and MDA-MB-231 (right panels) xenograft tumors in athymic female mice treated with C75 in the absence or presence of estrogen (E_2_) or tamoxifen. The maximum length for each treatment was 50 days. Results are presented as the mean tumor volume ± S.D. (*n* = 10 mice/experimental group). n.s. not significant.

**Figure 7 cancers-13-01132-f007:**
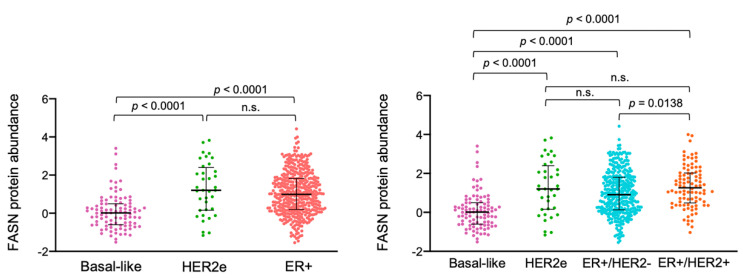
Differential enrichment of FASN protein expression in breast cancer subtypes. FASN protein expression levels in primary breast tumors from The Cancer Proteome Atlas (TCPA, http://tcpaportal.org, accessed on 30 October 2020) were classified into distinct breast cancer subtypes, namely basal-like (*n* = 95), HER2-enriched (*n* = 36), ER+ (*n* = 436), ER+/HER2− (*n* = 344), and ER+/HER2+ (*n* = 92). (ANOVA with Dunn’s multiple comparison test). n.s. not significant.

**Figure 8 cancers-13-01132-f008:**
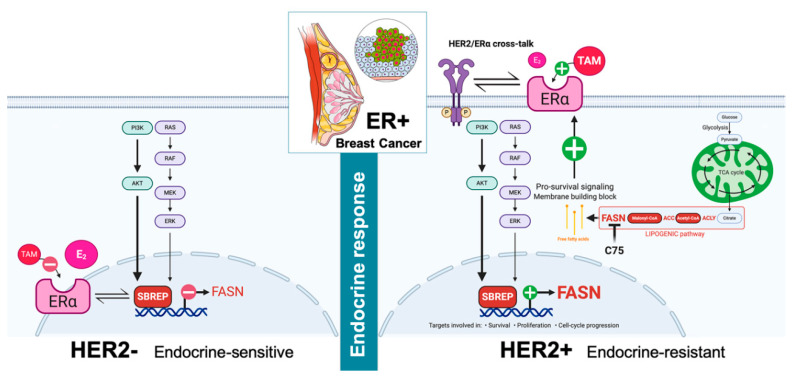
FASN enables HER2-driven breast cancer resistance to tamoxifen. The stimulatory effects of estradiol on FASN expression can be blunted by anti-estrogens in endocrine-responsive ER-positive/HER2- negative breast cancer cells (left). In ER positive/HER2- positive breast cancer cells, however, an AKT/MAPK-related constitutive hyperactivation of FASN expression becomes unaltered in response to estradiol and could be further enhanced by tamoxifen (right). FASN pro-survival signaling is co-opted by HER2 pathway activation to enable breast cancer resistance to tamoxifen. FASN inhibitors may be clinically relevant to countering resistance to tamoxifen in luminal B-like, ER+/HER2+ breast carcinomas. Created with BioRender.com and Mindthegraph.com.

## Data Availability

The data that support the findings of this study are available from the corresponding authors, upon reasonable request.

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
