# Peer review of "Fatty Acid Synthase Confers Tamoxifen Resistance to ER+/HER2+ Breast Cancer"

_cancers, 2021, doi:10.3390/cancers13051132_

Round 1

Reviewer 1 Report

Authors answer all my previous questions and they have introduced new experimental data as I requested.

Author Response

We would like to thank you for accepting our manuscript for publication in Cancers

Reviewer 2 Report

I am unsure if the new data warrants the change in title to “is a driver of”. The previous title was probably more adequate.

My initial comment about the relevance of the study in the field has not really been addressed. My concern was that all the references except one (21-26) at the bottom of p2 were from the first author. It makes difficult to judge the importance of the study.

Lines 110-111: review sentence as verb missing after “and/or”.

Lines 575-576: review fulvestrant source as it is not given as a pellet.

Author Response

Our deep interest in FASN as a target for therapeutic intervention stemmed from findings more a decade ago that tumor cells addicted to FASN-driven lipid signaling show significantly reduced growth and viability upon FASN inhibition (refs. 21-26). Since then, however, we and others have been unable to resolve the apparent discrepancy between the basic science-discovery bench aspects of FASN blockade and the awaited bedside effects of clinical-grade FASN inhibitors (FASNis). The demonstration of target engagement and early signs of clinical activity with the first orally available, selective, potent and reversible next-generation FASNi developed by Saginet Biosciences (formerly 3-V Biosciences) has reignited interest in FASN as a well-credentialed target for drug development. We honestly believe that the relevance of the current study revealing that FASN could be involved as an intrinsic factor of the transition to endocrine resistance in ER+/HER2+ breast cancer is that provides strong rationale for incorporating next-generation FASN inhibitors (e.g., TVB-2640) in the therapeutic arsenal against de novo endocrine resistance in patients with ER+/HER2+ luminal B-like breast cancer.

The title of the manuscript has been changed to:

Fatty Acid Synthase Confers Tamoxifen Resistance to ER+/HER2+ Breast Cancer

We have now revised sentences/verb usage in Lines 110-111:

"either increased the number of cells expressing FASN and/or increased the levels of FASN expression in individual cells"

Thank you for pointing out our inadvertent mistake regarding fulvestrant source.

Lines 575-576:

"...single-agent fulvestrant (5 mg/100 uL s.c. injection twice a week; Tocris Bioscience), ..."

Reviewer 3 Report

The revised version of this manuscript is acceptable for publication. 

Author Response

We would like to thank you for accepting our manuscript for publication in Cancers

This manuscript is a resubmission of an earlier submission. The following is a list of the peer review reports and author responses from that submission.

Round 1

Reviewer 1 Report

This is an interesting article, in which the expression and/or activity of FASN is associated with the acquisition of resistance to endocrine therapy with tamoxifen in breast cancer.

Initally, in MCF-7 cell line a model of luminal A breast tumor, the authors demonstrate that treatment with estradiol increases the expression of FASN and this increase is blocked by Tamoxifen, suggesting a genomic effect of estradiol, mediated by its receptor. The effect of estradiol on FSNA expression appears to be transcriptional, since a reporting gene under the control of a minimal 187 bp FASN promotor, mimics the results obtained with estradiol and tamoxifen. interestingly the effect seems to be mediated by an SREBP-binding site, apparently regulated through the PI3K/AKT pathway. Similar results are obtained in the T47D line, another model of luminal-A breast tumor. Interestingly, when the experiments are conducted in the MCF-7/HER-18 line, which has been transfected to overexpress HER 2, estradiol maintains the ability to promote tumors in immunosuppressed mice and to positively regulate the expression of FASN, although in this case tamoxifen does not antagonize estradiol activity, suggesting that these cells are resistant to tamoxifen therapy.

The results in MCF-7/HER18 a cell line transformed into a kind of luminal line B by the expression of HER2, are reproduced in line BT-474, a genuine luminal cell line B. On the other hand, negative control of the MDA-MB-231, a triple negative cell line, also behaves as expected.

Subsequent studies in patients with luminal A and luminal B tumors, show that high expression of FASN is a prognostic factor of low survival in luminal B tumors, which often have resistance to tamoxifen therapy, suggesting that the hypothesis of the authors that FASN is involved in the acquisition of tamoxifen resistance, in tumors expressing HER2, is correct. That is also supported by in vitro experiments with the minimum FASN promoter and by experiments with mice, since the FASN inhibitor, C75 reverses the resistance to tamoxifen. Although the article is well written and the experiments support the hypothesis, there are however some points that would require additional controls.

-It is difficult for me to assume that a minimal 187 pb construction of the promoter perfectly match what happens in the actual promoter, since we cannot be sure that chromatin and the distribution of nucleosomes in the real promoter, is similar in the reporting gene and also because usually transcription is affected by the cellular contest. Only one FASN expression data is included in the article in Figure 1. In my opinion, it would reinforce the article's line of argument to isolate RNA in control cells and in those treated with different agents and, to determine the expression of FASN mRNA, demonstrating that the data obtained with luciferase, correlate or not with the activity data of the endogenous promoter in the same treatments and cellular models.

-A second control to avoid the possibility that off-targets effects of the C75 inhibitor (FASN activity inhibitor) were responsible for the results obtained will be to eliminate, at least in one of the model used, FASN expression using a specific siRNA or similar and to demonstrate that the loss of FASN expression mimics the effect of the inhibition of FASN activity achieved with C75.

-Finally, I have a slight discrepancy in the interpretation of the result with inhibitor U0126 in Figure 1B. The authors conclude that estradiol loses its activator effect in the presence of U0126, but this is not what I see. In the presence of UO126 basal activity of the reporter gene decreases in a statistically significant way, when estradiol is present the basal activity is identical at the activity observed in control cells, so if we compare estradiol plus UO126 with UO126 alone, there is a positive effect of estradiol.

Reviewer 2 Report

This paper investigates the role of FASN in HER-2 driven tamoxifen resistance of ER+ breast cancer cell line. There is an interesting and well conducted set of in vitro and in vivo experiments that uncover a potential therapeutic intervention (C75 inhibitor of FASN).

It is difficult to evaluate the relevance of the findings in the field (introduction, bottom p2) as a lot of the reference are from the authors. A description of C75 and its importance in breast cancer (or cancer in general) is necessary. What is its relevance in the field?

In the present state of the paper, Figure 6 and associated text should be removed as it is very speculative and seem to have been added to bring some clinical relevance. The split between Luminal-A and -B ER+ (while real in the clinic) is not based on any results presented here. If the authors want to include this data, they need to bring additional data demonstrating a difference between the two types (is FASN expressed in more cell/at a higher level in Luminal-B tumors by IHC? Are there datasets that could show differential expression between the two types? …). What happens in other tumour types (TNBC, HER2 amplified?

Minor comments:

Lane 112: The article might gain in showing the WB

Fig 1A needs scale bars.

Fig 2, top: move the 6th bar before the 8th and delete the 5th and 7th which are duplicates (of 1st and 11th). That would simplify the figure.

Doses and regimen of tamoxifen and fulvestrant need to be indicated.

Reviewer 3 Report

In this paper, the authors identify FASN as a novel mediator of endocrine resistance in luminal B like, ER+/HER2+ breast cancer. FASN might be exploited as potential therapeutic target and a marker for Tamoxifen resistance. 

Here are some major points that should be addressed before the manuscript is accepted:

  1. This study is a follow up from the group's previous paper (Corominas-Faja et al, 2917; PMC5784426) that also addresses the importance of FASN in HER2-driven breast cancer. However, reference to this paper is buried in the introduction and not discussed much in this current manuscript. Please address why FASN did not distinguish between good and bad prognosis in Figure 2&3 in Corominas-Faja et al for luminal B but did so in this study? Were the patient cohort treated with Tamoxifen in the previous study? Please cite this paper and clarify any discrepancies and the cohorts used in the current study. 
  2. In section 4.6, what are the doses and mode of administration for Tamoxifen or Fulvestrant used in the in vivo study? These are critical information that should be included for the sake of scientific rigor and reproducibility. 
  3. What is the amount of estrogen given to the mice? This information should be included for scientific rigor and reproducibility. 
  4. On Line #280, is it said that ER+, HER2-, Ki67-low and luminal-B ER+, HER2-, Ki67-high are considered for the survival analysis. Luminal-B type breast cancer can be both HER2+ or HER2-. Does FASN distinguish between good and bad prognosis within luminal B that is HER2+?
  5. Classification of luminal-B breast cancer needs to be clarified throughout the paper, e.g., line #287.
  6. What is the ER protein levels in MCF7/HER2-18 cells?
  7. Does the ER protein levels change with C75 treatment in cells and tumors? A western blot analysis of ER in the cells or IHC treated tumor samples will be informative.